# Age-Appropriate Advance Care Planning in Children Diagnosed with a Life-Limiting Condition: A Systematic Review

**DOI:** 10.3390/children9060830

**Published:** 2022-06-03

**Authors:** Julie Brunetta, Jurrianne Fahner, Monique Legemaat, Esther van den Bergh, Koen Krommenhoek, Kyra Prinsze, Marijke Kars, Erna Michiels

**Affiliations:** 1Julius Center for Health Sciences and Primary Care, University Medical Center Utrecht, 3584 CG Utrecht, The Netherlands; j.brunetta@students.uu.nl (J.B.); m.c.kars@umcutrecht.nl (M.K.); 2Division of Pediatrics, Wilhelmina Children’s Hospital, 3584 EA Utrecht, The Netherlands; 3Princess Máxima Center for Pediatric Oncology, 3584 CS Utrecht, The Netherlands; m.m.legemaat@prinsesmaximacentrum.nl (M.L.); e.m.m.vandenbergh@prinsesmaximacentrum.nl (E.v.d.B.); k.b.krommenhoek@students.uu.nl (K.K.); k.j.prinsze2@students.uu.nl (K.P.); e.michiels@prinsesmaximacentrum.nl (E.M.)

**Keywords:** palliative care, life-limiting conditions, pediatrics, adolescents, advance care planning, age-appropriate, development, cognitive functions, young adults, interventions

## Abstract

Pediatric advance care planning (pACP) is an important strategy to support patient-centered care. It is known to be difficult, yet paramount, to involve the child in pACP while adjusting treatment to age and the corresponding stage of development. This systematic review was aimed to evaluate the age appropriateness of pACP interventions by assessing their characteristics, content, and evidence. CINAHL, Embase and MEDLINE were searched from 1 January 1998 to 31 August 2020 in order to identify peer-reviewed articles containing strategies and tools to facilitate pACP in both children (0–18 years) with life-limiting conditions and their families. An assessment of quality was performed using Cochrane tools and COREQ. The full protocol is available as PROSPERO CRD42020152243. Thirty-one articles describing 18 unique pACP tools were included. Most tools were developed for adolescents and young adults. In most cases, the interventions tried to assess the child’s and family’s preferences concerning their current and future hopes, wishes, and goals of the care. This was aimed to enhance communication about these preferences between children, their families, and health-care providers and to improve engagement in pACP. The relevance of an age-appropriate approach was mentioned in most articles, but this was mainly implicit. Seven articles implemented age-appropriate elements. Six factors influencing age appropriateness were identified. Tools to support pACP integrated age-appropriate elements to a very limited extent. They mainly focused on adolescents. The involvement of children of all ages may need a more comprehensive approach.

## 1. Introduction

Children with life-limiting conditions often receive highly complex care over a long period of time. This care may include high-risk treatments with severe side effects and palliative care services. The medical conditions and care needs of these children often interfere with their daily life, including their social activities [1]. These children live with the burden of invasive treatment procedures, hospital admissions, and (often) side-effects from therapies [2,3]. However, these children are not routinely asked about their experiences regarding living with illness [3]. In addition, the child’s voice is not systematically included in decision making or when discussing treatment preferences [3]. The involvement of parents and children, in a way appropriate for both age and level of development [4], in decision making is considered obligatory in family-centered health care. However, involving children and families is challenging in practice. Uncertainty about prognoses, fears of disrupting coping strategies, intercultural differences, and the changing demands of developing children make clinicians feel reluctant to initiate conversations about future care with children and their families [5,6,7]. Furthermore, tools to support the participation of children in decision making regarding their own health care are scarce. However, these tools are needed if the child’s perspective when discussing goals and preferences for care and treatment is to be included [8]. In the literature, reports of pediatric advance care planning (pACP) interventions are increasing [9,10,11,12,13]. They are intended to connect the expertise of the child and family with that of the medical team in order to define the shared goals of their care and to better communicate these to all caregivers involved in the child’s treatment [14,15,16]. Advance care planning includes considering the voice of the child, either by listening to the child itself or by identifying the child’s perspective through the parents or other involved caregivers. An age-appropriate approach is needed to identify the child’s perspective in an adequate way. The child’s age, with their corresponding cognitive development, will influence their ability to participate in conversations [4]. The development age of children was divided by Piaget into four stages based on the level of development adequate for the calendar age. These stages are: the sensorimotor stage (from zero to two years old), the pre-operational stage (from two to seven years old), the concrete operational stage (from seven to 11 years old), and the formal operation stage (12 years and above) [5,17]. These stages were based on the idea that different age groups correspond with different levels of cognitive development. However, due to multiple factors, including illness, development age can differ from calendar age. We refer to age appropriateness for a certain development age. Clinicians experience difficulties in incorporating age-appropriate communication strategies tailored to individual needs of children [5]. Even when a child is not too young or cognitively impaired and is able to participate in a conversation, clinicians still tend to focus on the parents when discussing the child’s illness [18]. While parents will act as advocates for their child’s health, their needs, interests, and coping strategies may interfere with the child’s perspective and best interests. This limits the parents’ ability to discuss or represent the voice of their child, particularly during the palliative phase [19,20]. Although the outcomes of pACP interventions are promising, it is unknown to what extent interventions elicit the voice of the child in a manner appropriate to their age. The adequate participation of children in ACP can therefore be achieved by using strategies that consider the development ages of children. An overview of pACP interventions appropriate to different ages would be helpful in order to gain insight into strategies for adequately involving children with life-limiting conditions in their own health-care decisions. To our knowledge, an overview of such pACP interventions is still lacking. Therefore, this review was aimed to identify if and how pACP interventions incorporate elements appropriate to the child’s age.

## 2. Materials and Methods

### 2.1. Data Sources and Searches

This review was structured using The Preferred Reporting Items for Systematic Reviews and Meta-analyses (PRISMA) checklist and the Palliative Care Literature Review Iterative Method (PALETTE) [21,22]. A structured computerized literature search was conducted in three databases: CINAHL, Embase, and MEDLINE. The search strategy was developed in collaboration with an information specialist and included terms describing the following domains: advance care planning, critical illness, and pediatrics (Table 1). These terms were searched for in all fields, with synonyms and truncations added. Three reviewers independently screened all abstracts in order to select papers reporting on pACP tools in children (0–18 years old) with life-limiting conditions [23]. We resolved questions about whether to include some papers through discussion. The reference lists of studies we included were hand-searched for additional relevant articles.

### 2.2. Study Selection

Articles published in English in peer-reviewed journals between 1 January 1998 and 31 August 2020 were eligible for inclusion if they reported on a well-described strategy or tool for supporting pACP. pACP was defined as a strategy to identify preferences and goals for future care and treatment [24] by connecting the expertise of the child and family with the expertise of the medical team [14,15,16]. Exclusion criteria were systematic reviews, articles published before 1998, and articles reporting on prenatal advance care planning. The full texts of potentially eligible studies were independently assessed by three reviewers. Disagreements were resolved in discussion with members of the research team. If an article did not provide a comprehensive description of the tool, then more detailed information was requested from the first author by email.

### 2.3. Data Extraction and Quality Assessment

Data extraction was conducted by two authors using a predesigned form [25]. Data regarding the content of the tool, the person conducting the conversation about ACP, its target population, and the items and outcomes related to age appropriateness were extracted. Two authors independently evaluated the studies’ methodological rigor by using the appropriate tool. Disagreements were resolved through discussion. We used the Cochrane Collaboration’s risk of bias tool for randomized controlled trials [26]. This enabled us to evaluate the following: random sequence generation, allocation concealment, the blinding of participants, the blinding of outcome assessments, incomplete outcome data, and selective reporting. One or zero points were allocated when there was a low or high bias risk, respectively. An unclear risk of bias was noted with a question mark, resulting in zero points. A total score of six was achievable. Observational studies were evaluated with an adapted version of the Cochrane bias tool. This enabled us to appraise the selection of study population, the comparability of study groups, the standardization of intervention protocols, the standardization of outcome measurements, any missing data, any confounders, and any selective outcome reporting [26]. Points were assigned as mentioned above. A total score ranging from zero to seven was counted. Qualitative studies were evaluated using the COmprehensive consolidated criteria for REporting Qualitative research (COREQ), assessing 32 criteria concerning three domains: the research team and reflexivity, the study design, and the analysis and findings [27]. Scores of one, 0.5, and zero points were assigned when the score was, respectively, properly described in the manuscript, incomplete, and not described. Assessments of both the risk of bias and the quality of reporting were conducted for mixed-method study designs. A few articles were not critically appraised due to their narrative, non-empirical study design. This review was exploratory in nature, so inclusion was not affected by the quality of selected papers [28].

### 2.4. Data Synthesis and Analysis

The researchers listed the characteristics and content of the pACP tools and their reported empirical outcomes. A narrative synthesis was provided to summarize the results [29]. Any age-appropriate elements and related theoretical groundings were identified by using a qualitative approach. Age-appropriate elements were defined as components of the tool that were adapted to a specific age and corresponding stage of cognitive development. It was reported whether elements were adapted on the basis of age groups in general or, specifically, on the development capacities that matched a specific stage of development. Fragments of articles related to age appropriateness were extracted. The open coding of these fragments resulted in a list of codes related to age appropriateness. Overarching concepts that describe factors influencing age appropriateness in the context of pACP were identified [30]. The protocol of this review is registered in the public registry PROSPERO, with registration number CRD42020152243.

## 3. Results

The search identified 11,685 unique hits, resulting in 62 articles eligible for full-text screening. Thirty-four articles were excluded after full-text screening. Twenty-seven had no description of a pACP tool, one article reported on adults, and six articles were excluded based on their study design (systematic review, prenatal pACP or published before 1998). Thirty-one articles, reporting on 18 unique pACP tools, met the inclusion criteria (Figure 1). Twenty-one articles were original empirical studies reporting outcome data, including six trials [31,32,33,34,35,36], six observational studies [37,38,39,40,41,42], four qualitative studies [11,43,44,45], and five studies that used mixed methods (observational and qualitative study design) [46,47,48,49,50]. Ten articles described a tool or intervention without reporting any empirical data [51,52,53,54,55,56,57,58,59,60]. Most studies (*n* = 24) were conducted and published in the USA [31,32,33,34,35,36,37,38,39,40,41,42,43,46,47,48,49,51,52,53,56,58,59,60], five were published and conducted in the UK [44,45,50,54,55], one was published and conducted in the Netherlands [11], and one was published and conducted in Canada [57].

### 3.1. Risk of Bias and Quality of Reporting

Table A1, Table A2 and Table A3 of Appendix A show an overview of the scores per article, with regard to the risk of bias and an assessment of the quality of reporting. The total scores per study are presented below (Table 2, Table 3, Table 4, Table 5 and Table 6: Article Characteristics). The six randomized controlled trials had a median total score of 4 out of 6 (range: 3–5). All six articles could not blind their participants and therefore did not meet this criterion. For observational studies (*n* = 6) and the quantitative parts of mixed-method studies (*n* = 5), the median scores were 3 (range: 2–5) and 4 (range: 2–5), respectively. The qualitative studies (*n* = 4) had a median total score of 10 out of 32 (range: 4–18). For mixed-method studies (*n* = 5), the median total score of the qualitative part was 8.5 (range: 4.5–12).

### 3.2. Intervention Characteristics

Table 7 presents an overview of the characteristics of the 18 pACP tools. Most interventions focused on conversations with children and their parents or surrogates as a key element of ACP [11,31,33,34,35,36,37,39,42,43,44,45,46,47,50,52,53,54,55,57,58,60]. Seven articles were only concerned with patients [40,48,49,51,56,59]. Two interventions targeted parents of children [38,41,47]. Some tools were used to study specific disease groups, such as oncology (*n* = 6) [35,36,37,38,39,41,42,43,46,52,58,59] and HIV/AIDS (*n* = 1) [31,32,33,34,53], whilst most tools focused on children with life-limiting conditions in general (*n* = 12) [11,40,44,45,47,48,49,50,51,54,55,56,57,60]. Twenty-five articles specified their research population’s age, ranging from zero to 28 years. Figure 2 displays the children’s ages of the target population per article. The authors of one article studied children from the age of 13 years [43]. A few studies did not specify the age of the child [44,54,55,56,58,59]. Nine studies researched children of all ages, including young adults [11,37,38,40,41,45,47,50,60]. Most articles were focused on adolescents and young adults [11,16,31,32,33,34,35,36,37,38,39,40,41,42,43,45,48,49,50,52,53,60], and only a few included young children [11,37,38,40,41,45,46,47,50,57,60]. Three studies described a specific age in their research population but did not explain their choice of this age [46,51,57]. Among those intervening in the care were a broad diversity of clinicians including pediatricians, nurses, clinicians, and unspecified certified facilitators. Conversation topics included: disclosing hopes, wishes, goals (of care), preferences for care and treatment, family and patient needs, and the planning of future or end-of-life care. The ACP was approached as a longitudinal face-to-face process with multiple conversations. Most articles did not specify the race or ethnicity of their target population [11,40,43,44,45,50,51,52,54,55,56,57,58,59,60]. The most common population backgrounds were Caucasian [35,36,38,41,42,46,47,49] and African American [31,32,33,34,39,48,53]. The importance of a culturally appropriate pACP intervention was mentioned in most articles (*n* = 20) [11,31,32,33,34,35,36,42,44,45,47,48,50,52,53,54,55,56,58,60].

### 3.3. Attention to Age Appropriateness

Age-appropriate characteristics are summarized in Table 8. The concept of an age-appropriate approach was mentioned in two thirds of the articles [11,32,33,35,36,37,42,44,48,49,50,51,52,53,54,55,56,57,58,59,60] in an implicit or explicit way. However, no clear definition of an age-appropriate approach to pACP was described. Seventeen articles mentioned the age-appropriate concept in an implicit way without linking the importance of adapting the tools to the development of the child [35,36,37,40,42,44,48,49,50,51,52,53,54,55,57,58,60]. An example of an implicit description of the age-appropriate concept is cited in Box 1.

Box 1Example of implicit description of the concept of age appropriateness.“Most adolescents aged 14 years and older do not differ from adults in their capacity to make informed treatment decisions, and their understanding of death is no less mature than that of adults” [42] (p. 2).

Few articles referred to the concept in an explicit way by describing any implications of using the concept [11,32,33,56,59]. An explicit description of the age-appropriate concept is presented in Box 2.

Box 2Example of explicit description of the concept of age appropriateness.“Developmentally, the AYA period is characterized by emerging abstract thinking and an evolving sense of vulnerability. Given this complex developmental stage, AYA patients may benefit from the use of specialized tools to facilitate abstract consideration of factors involved in decision making” [59] (p. 2).

Although most articles referred to age appropriateness as a concept in some way, this was generally not translated into specific elements of the described tools nor specified for different levels of development. Twelve articles provided general recommendations to implement age appropriateness in pACP tools [11,32,37,48,49,50,51,54,55,56,58,59]. Fourteen studies claimed their tools to be age-appropriate [11,32,33,36,48,49,50,51,52,53,56,57,59,60], yet only seven articles implemented elements adjusted to the age of their population. Most of these articles adapted the language to the child’s age [11,48,50,51,56,57], used age-appropriate images [49,50], or added a glossary [48,56]. One article referred to their pACP guide as containing family-centered language [57]. These elements mainly focused on adolescents and young adults [48,49,51] or did not specify a particular age of their target population [56,57]. None of the articles explained why these adaptations meet the development needs or capacities of studied children nor explored the development needs of children in general.

Another way to contribute to the age-appropriate concept was by evaluating a tool for its age appropriateness. Most articles did not report any empirical study data regarding the age appropriateness of the tool used by participants. Twenty-eight studies described or evaluated the effectiveness and the child and family preferences of their tools, but none of them specifically evaluated their age appropriateness [11,31,32,33,34,35,36,37,38,39,41,42,43,44,45,46,47,50,51,52,53,54,55,56,57,58,59,60]. Only three studies examined the age appropriateness of their tool by asking adolescents and young adults if the tool was considered appropriate for themselves and other participants of their age [40,48,49]. The development stage or capacities to participate in the pACP of the children were not described or researched. None of the articles examined age appropriateness in young children. Age-appropriate outcomes were reported by providers in one article [40] and by AYAs in three articles [40,48,49]. These studies showed that AYAs considered pACP tools to be age-appropriate [40,49] and could be introduced before the age of 18 [40]. AYAs experienced the tools as helpful [49]. Only one article examined the perspective of the providers, revealing that pACP tools were considered less appropriate for AYAs and therefore contradicting AYAs’ opinion on age appropriateness. About half of the providers reported thinking that pACP conversations should occur after the age of 18 [40].

### 3.4. Factors Influencing Age Appropriateness

We identified four factors related to age appropriateness that might influence the pACP approach: willingness to participate, ability to participate, social identity, and legal responsibilities. How these factors function at certain development stages was not clearly described. Table 9 shows an overview of these factors per article. Articles were marked with an ‘x’ when contributing to this factor.

Sixteen articles stated that children, especially AYAs, show a willingness to participate in pACP conversations [11,31,32,33,34,35,36,39,40,42,43,48,51,53,56,59]. Articles explored the child’s willingness to participate by asking this to the children themselves and their parents. Willingness reflects the motivation of the child to be involved in a pACP conversation and clarifies to what extent this may be so. Many articles cited previous research on this subject, which showed that adolescents and young adults have a desire to participate in pACP. Few declared the same desire among young children and teenagers [11,34,43,56].

Another factor we identified was the ability to participate in pACP. This was referred to by three different sub-themes. Firstly, multiple articles reported that children and adolescents are cognitively able and sufficiently mature to make decisions, medical or otherwise, and that they therefore should be involved in pACP [11,32,33,35,36,42,44,48,49,51,53,54,55,56,58,60]. Most articles did not specify which cognitive capacities are needed but described cognitive capacities in general. The second sub-theme was the understanding of how a child’s own disease process contributes to their participation in pACP. This may indicate whether or not they are able to understand the content of a pACP conversation [33,35,36,42,48,49,51,53,55,56,57,58,59]. The subjects we identified were an understanding of the consequences of decision making [33,35,53,56,58], medical concepts (health, illness, death) [36,42,48,51,55,57,59], and an understanding of treatment decisions in general [36,55,56,58]. The final sub-theme was cognitive impairment. Many articles excluded patients with cognitive impairment because they experienced this as limiting or complicating age-appropriate pACP [31,32,33,34,35,36,39,42,43,52,53]. However, separate from decision-making capacity and understanding, cognitive impairment was identified as a factor on its own that influences the ability of a child to participate in conversations.

Five articles described a developing social identity in adolescence as a factor related to age appropriateness [32,48,51,55,56]. During adolescence, children develop an awareness of themselves and others, which influences children’s preferences and goals in pACP.

Some articles described the law requesting an advance directive, or living will, starting from a certain age [31,33,34,38,39,48,49,53,56]. These legal documents or conversations were sometimes described as part of the pACP conversations. Laws determine what is considered a legal age in participating in own health-care decisions. In some articles, younger age groups (18 years old or younger) were excluded from such topics or conversations [31,33,34,38,39,48,53,56].

## 4. Discussion

To our knowledge, this is the first systematic review examining age-appropriate characteristics and outcomes in pACP interventions for children with life-limiting conditions. Thirty-one articles reporting on 18 unique pACP tools were identified. Although pACP is aimed to emphasize the preferences and goals of children and their parents, the voices of children are explored by the interventions to a very limited extent. Two thirds of the studied articles referred to the age-appropriate concept; however, none of the studies comprehensively examined the development stage of their target population. Few interventions contained elements adjusted to the development of the child, or evaluated the intervention on age appropriateness [40,48,49]. The factors contributing to age appropriateness identified from the studies we investigated were: willingness to participate, ability to participate, developing a social identity, and legal responsibilities.

### 4.1. Defining Age Appropriateness in pACP

In this review, we have defined age appropriateness as the level of cognitive development of a child corresponding to a certain age. Cognitive development can differ between individuals of the same age and can fluctuate in children with life-limiting conditions [61,62,63]. Age-appropriate pACP tools would therefore benefit from adjustments to the development stage of a child with a life-limiting illness.

In this review, we identified different factors (willingness to participate, ability to participate, legal responsibilities and social identity) that might characterize or influence the development stage. Piaget described different stages in cognitive development in children [17]. The literature is not, however, clear about whether these stages could be used for children with life-limiting illnesses. However, it does provide general information on the development capacities and the comprehension of topics related to ACP. As ACP is intended to be used to discuss future care preferences, children might benefit from an understanding of the medical concepts involved and also from a greater role in medical decision making. A review of medical decision making in children and adolescents showed that four cognitive capacities are needed: communicating a choice, understanding, reasoning, and appreciation [64]. In this way, the development stage of the child, with corresponding cognitive capacities, determines their ability in medical decision making. Expressing a choice, the first criterion, can either be accomplished via language or non-verbal communication [65,66]. Starting from the age of five years, children have a proper understanding of language, and this is a first step towards medical decision making. Non-verbal communication helps in assessing a child’s preferences but is excluded as legal consent [67]. The second criterion, understanding information on one’s own medical treatment, requires different neurological capacities in decision making [65,66]. Previous studies have shown that children from the ages of seven to ten years can orient and maintain attention [68,69,70], those from six to 12 years old can memorize [71,72], and those from the age of ten years old can recall received information [73,74,75]. Aside from understanding information on treatment, the comprehension of understanding of concepts such as illness, life, and death depends on the cognitive understanding of death as a biological act [76,77,78,79,80]. This can be fully understood starting from the ages of five to seven years old [78,79,80,81]. The understanding of sub-concepts such as irreversibility, universality, personal mortality, inevitability, causality, and unpredictability might even begin at the age of three [77,78,79,80,81]. The articles we researched provided some basis to this theory, implying that children, starting from a young age, should not be excluded from these topics in pACP. Children from the ages of six to eight years can logically reason [82,83] about decision-making consequences, including risks and benefits, which is the third criterion [65,66,84]. This capacity further develops in adolescence, therefore meaning that they can understand more complex issues [82]. Few of the articles we researched mentioned that children can understand the consequences of decisions, indicating that children are able to reason regarding logically their own pACP decisions and should therefore be included in weighing different treatment options. The last criterion, appreciation, indicates that children from the ages of three to four years start recognizing their own norms and values, as well as the effect of these on their own life [67,85,86]. This implies that preferences and hopes in pACP could be explored in a more simple manner and early in childhood. Most studied articles focused on pACP interventions for older children and may have underestimated the value for younger children. Adolescence is considered an interesting development phase in decision making in which children develop a social identity and awareness of themselves and their peers [87]. They highly value the acceptance of peers, which influences decision making [88]. Adolescents make more decisions offering swift rewards in the presence of the other peers [89]. Altogether, Grootens-Wiegers et al. stated that children from the age of 12 are expected to be competent in decision making [64]. Legally, children from the age of 12 years are allowed to make joint decisions on medical issues with their parents. From the age of 16, they can make decisions on their own [90]. In the USA starting from the age of 18 they are allowed to give informed consent for participating in clinical trials [91]. pACP can play a valuable role in preparing children for decision making. However, ACP was not developed for contemporaneous medical decision making; rather, it was developed for preparing certain decisions in the future. In this way, children can participate and have a feeling of control in their own disease process. This can only be achieved when the child’s level of development is assessed as part of pACP or prior to the initiation of pACP.

We identified cognitive impairment as a factor influencing the concept of development. Approximately half of the children with life-limiting illnesses also suffer a degree of cognitive impairment [92]. Multiple articles excluded children with cognitive impairment, indicating that it could complicate participation in pACP. Cognitive impairment is defined as a deficiency in cognitive function consisting of multiple capacities: memory, general intelligence, learning new things, language, orientation, perception, attention, and concentration and/or judgment [93]. Cognitive impairment is a broadly used term in which one or more cognitive functions are affected in general. Engaging these children in pACP would therefore benefit from adaptations to the development of their cognitive functions.

Most pACP articles on children with life-limiting conditions reported that pACP interventions need to be culturally appropriate. Preferences in discussing pACP topics differ between cultures [94]. However, the literature is not clear what is considered age-appropriate pACP participation in different cultures. Cognitive development, and therefore level of the child’s participation in conversations and decision making, is influenced by cultural differences that affect parenting roles, government guidelines and education [95]. To involve children in pACP in an adequate and age-appropriate manner, their ethnical background should be considered.

Even though defining different stages in the development of children with life-limiting illnesses would be helpful, evidence indicates that the development, cognitive or otherwise, of a child is an individual, fluctuating, culturally-dependent and differentiating process with regard to different topics [61,62,63]. The development of a child can progress or regress individually during the life or disease process. When a child deteriorates, cognitive development can regress and changes in level of development and rate of assent could appear [90]. On the other hand, clinical experience has shown that children, especially adolescents, with life-limiting illnesses often seem to have a better understanding of death and dying compared to healthy children of their age [61]. This can be due to their greater experience with death than other children of their age [62]. However, the literature is inconsistent. Experiencing death through media [62] and what parents teach their children about biology and natural processes [63] stimulates children with regard to the concept of death and dying [62,63]. Diverse ACP topics might have different levels of development in one individual. A comprehension of life and death is only one aspect of the topics discussed in pACP. Other topics, such as preferences and hopes in general, could be easier for children to talk about and could be comprehended on another level of development. The comprehension of diverse topics in pACP differs in the development in children and should therefore determine their level of participation in that topic. The comprehension of one’s own body, for example, develops between the ages of four to six years [96], while the understanding of death develops later, between the ages of five and seven years [62,78,79,80,81].

### 4.2. Recommendations for Future Research

Development is an individual, fluctuating, culturally dependent, and differentiating process between topics and cognitive functions. The stage of development, rather than age, therefore gives direction in how to appropriately engage children with regard to their development. Research is needed to identify the specific characteristics of each development stage for children with life-limiting conditions. Elements that depend upon developments can then be determined and incorporated to create age-appropriate pACP interventions. Most pACP tools lack a comprehensive inclusion or description of age-appropriate pACP elements. Current literature on pACP does not provide sufficient insight in characteristics of age-appropriate elements. This complicates the appraisal of the usefulness of current tools. Two interventions were evaluated on age appropriateness by their target population of AYAs and their providers [40,48,49]. They were considered acceptable and useful for this age group. Age-appropriate elements, such as language and images, were incorporated, but an explanation of why these elements met the child’s developmental needs and capacities was lacking. Examining the intervention by the target population provides a first indication of the level of age appropriateness of a tool. However, more insight is needed regarding which development characteristics apply to certain age groups that are relevant in pACP. In this way, a framework of age-appropriate pACP elements can be designed and incorporated into existing and newly developed interventions. pACP tools can be ranked for their level of age appropriateness, which might be an indicator of high-quality family-centered tools. We were not able to apply any qualification to the level of age appropriateness of the pACP tools studied in this review without such a framework.

Creating and examining different stages in development for children with life-limiting illnesses would benefit from other fields of expertise. Experts in developmental psychology, even children themselves and their families, could determine what information or elements are considered age-appropriate [40,48,49]. The development stage should be frequently assessed because it fluctuates, which may require a separate tool. Creating an intervention that determines the level of development of a child regarding pACP topics could provide an indication of how clinicians can involve children in pACP conversations, e.g., which topics providers have to raise or questions to ask. The same fluctuating cognitive phenomena have been observed in dementia patients [97,98] in which cognitive capacities for ACP participation were assessed [97,99]. The literature shows that even patients with severe dementia can still share their preferences or wishes on a certain level [99]. Therefore, we should always explore children’s cognitive development and involve children in their own disease processes on a level that corresponds with their cognitive capacities.

Future research should investigate what is considered age-appropriate participation in different cultures, which has not yet been described in the literature. In creating an age-appropriate intervention, cultural norms and values should be incorporated to involve children in a way that is appropriate for their developmental stage.

### 4.3. Strengths and Limitations

This systematic review contributes to the body of knowledge of the young and evolving field of pACP. The research team assessed the content and characteristics of different pACP tools regarding age appropriateness. A team offering broad expertise evaluated the different aspects of age appropriateness. This review comprises the first steps towards the incorporation of age appropriateness in pACP. The results show that age appropriateness is considered important; however, the comprehensive elaboration of this concept is still in its infancy. These findings limit the opportunities for clinical implications for current practice while emphasizing the need for ongoing research to be able to develop a comprehensive age-appropriate approach in pACP. In addition, the variety of different study designs complicates any comparison of the role of the separate intervention elements. pACP may be an upcoming field of expertise, but it is still relatively new in advance care planning and there might be more influencing factors than we have discovered. Other fields of expertise might contribute to these factors. Most reviewed articles were published and conducted in the USA. This might limit the applicability of the findings to other countries. The USA articles were mainly dominated by one research group, which might have biased the results.

## 5. Conclusions

In this review, we have summarized the age appropriateness of existing pACP tools in children with life-limiting conditions. The relevance of an age-appropriate approach was mentioned in most articles, though mainly in an implicit way. None of the articles comprehensively examined the development stage of their target population. Four factors influencing age appropriateness were identified: willingness to participate, ability to participate, developing social identity, and legal responsibilities. Three articles evaluated their tools regarding age appropriateness. The tools integrated age-appropriate elements to a very limited extent, mainly focusing on adolescents and young adults. The involvement of children of all ages in pACP needs a more comprehensive approach.

## Figures and Tables

**Figure 1 children-09-00830-f001:**
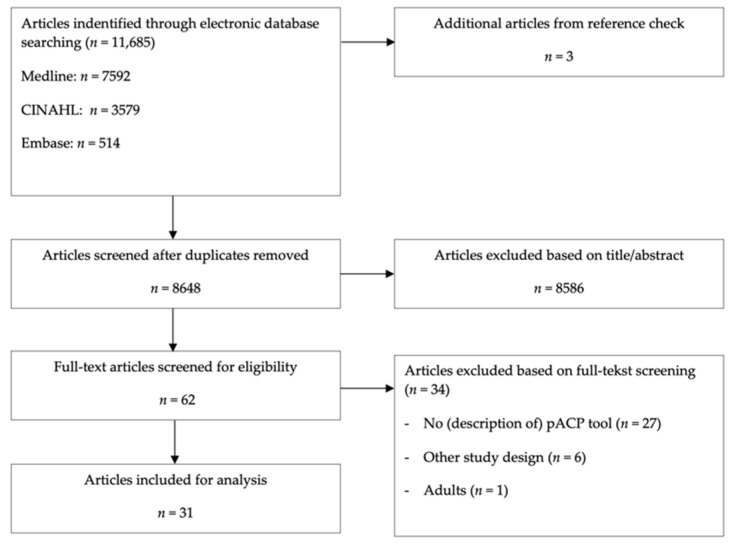
PRISMA flow diagram of the literature review process.

**Figure 2 children-09-00830-f002:**
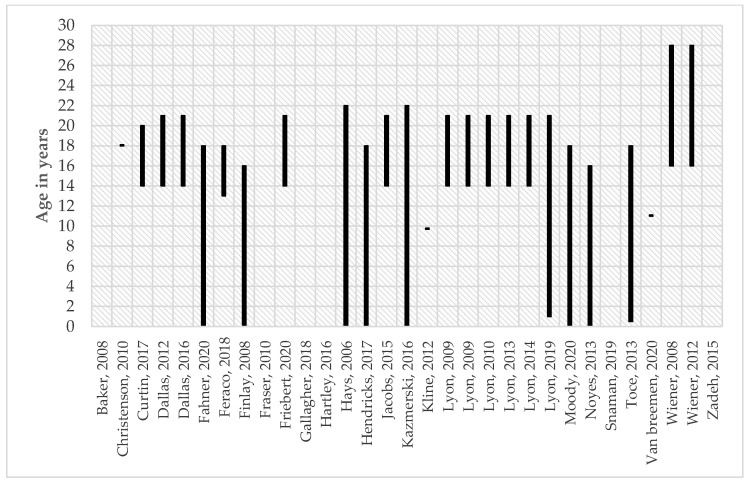
Age range in study population per article [11,31,32,33,34,35,36,37,38,39,40,41,42,43,44,45,46,47,48,49,50,51,52,53,54,55,56,57,58,59,60].

**Table 1 children-09-00830-t001:** Search string for Medline. Search date: 31 August 2020.

(critical illness[MeSH Terms] OR critical illness*[tiab] OR "critically ill"[tiab] OR life limiting condition*[tiab] OR life-limiting disease*[tiab] OR life threatening illness*[tiab] OR life limiting illness*[tiab] OR life threatening condition*[tiab] OR serious illness*[tiab] OR palliative care[MeSH] OR terminal care[MeSH] OR "palliative care"[tiab] OR "palliative medicine"[tiab] OR "palliative nursing"[tiab] OR "palliative period"[tiab] OR "palliative phase"[tiab] OR "palliative therapy"[tiab] OR palliative treatment*[tiab] OR "palliative supportive care"[tiab] OR "terminal care"[tiab] OR "terminal medicine"[tiab] OR "terminal period"[tiab] OR "terminal phase"[tiab] OR EOL[tiab] OR end of life*[tiab])
And
("advance care planning"[MeSH] OR "advance directives"[MeSH] OR "decision making"[MeSH] OR "living wills"[MeSH] OR "patient participation"[MeSH] OR advance care plan*[tiab] OR ACP[tiab] OR pACP[tiab] OR advance decision*[tiab] OR advance directive*[tiab] OR advance medical directive*[tiab] OR advance healthcare planning*[tiab] OR advance medical planning*[tiab] OR advance statement*[tiab] OR "do not hospitalize"[tiab] OR "do not hospitalise"[tiab] OR "do not resuscitate"[tiab] OR "do not attempt cardiopulmonary resuscitation"[tiab] OR "DNR order"[tiab] OR DNACPR[tiab] OR "planning ahead"[tiab] OR "refusal of treatment"[tiab] OR treatment limitation*[tiab] OR conversation guide*[tiab] OR guide*[tiab] OR program*[tiab] OR procedure*[tiab] OR practice*[tiab] OR treatment limiting*[tiab] OR shared decision*[tiab] OR "patient participation"[tiab] OR "patient involvement"[tiab] OR "child centered care"[tiab] OR "person centered care"[tiab] OR "patient centered care"[tiab])
And
(Infan*[tiab] OR toddler*[tiab] OR minor[tiab] OR minors*[tiab] OR boy[tiab] OR boys[tiab] OR boyfriend[tiab] OR boyfriends[tiab] OR boyhood[tiab] OR girl[tiab] OR girls[tiab] OR girlfriend[tiab] OR girlfriends[tiab] OR kid[tiab] OR kids[tiab] OR child[tiab] OR children*[tiab] OR schoolchild*[tiab] OR school child*[tiab] OR adolescen*[tiab] OR juvenil*[tiab] OR youth*[tiab] OR teen*[tiab] OR underage*[tiab] OR pubescen*[tiab] OR puberty[tiab] OR pediatrics[MESH] OR pediatric[tiab] OR pediatrics[tiab] OR paediatric[tiab] OR paediatrics[tiab] OR school[tiab] OR school*[tiab] OR prematur*[tiab] OR preterm*[tiab] OR youth[tiab] OR youths[tiab] OR teen[tiab] OR teens[tiab] OR teenager[tiab] OR youngster*[tiab] OR child[MeSH] OR neonat*[tiab] OR baby[tiab] OR babies[tiab] OR toddler*[tiab] OR newborn*[tiab] OR postneonat*[tiab] OR postnat*[tiab] OR perinat*[tiab] OR preschool*[tiab] OR suckling*[tiab] OR picu[tiab] OR nicu[tiab] OR neo-nat*[tiab] OR neonat*[tiab] OR premature*[tiab] OR postmature*[tiab] OR pre-mature*[tiab] OR post-mature*[tiab] OR preterm*[tiab] OR pre-term*[tiab] OR playgroup*[tiab] OR play-group*[tiab] OR playschool*[tiab] OR prepube*[tiab] OR preadolescen*[tiab] OR junior high*[tiab] OR highschool*[tiab] OR senior high[tiab] OR young people*[tiab])

* Truncations were added.

**Table 2 children-09-00830-t002:** Evidence from randomized controlled trials.

Author, Year, Country *	Aim	Population (Age in Years), *n*	Outcome Parameters	Risk of Bias Total Score (6)
Dallas, 2016, USA [31]	FACE (FAmily/Adolescent-CEntered Advance Care Planning) vs. Healthy Living Control Condition	Adolescents with HIV (14–21) and their family decision maker, dyads *n* = 97 (I: 48, C: 49)	FACE: Participant enrollment and attendance Satisfaction based on positive and negative experienced emotions (Satisfaction Questionnaire) Serious adverse event	5
Lyon, 2009, USA [32]	FACE vs. Healthy Living Control Condition	Adolescents with HIV/AIDS (14–21) and surrogate, dyads *n* = 38 (I: 20, C: 18)	FACE: Participant enrollment, attendance, and retention Data completeness Satisfaction based on positive and negative experienced emotions (Satisfaction Questionnaire)	3
Lyon, 2009, USA [33]	FACE vs. Healthy Living Control Condition	Adolescents with HIV/AIDS (14–21) and surrogate, dyads *n* = 38 (I: 18, C: 17)	FACE: Family congruence Adolescent decisional conflict Quality of communication	3
Lyon, 2010, USA [34]	FACE vs. Healthy Living Control Condition	Adolescents with HIV (14–21) and legal guardian, dyads *n* = 38 (I: 18, C: 17)	FACE: Data completeness Psychological effects (based on anxiety and depression scales) Quality of life Physical effects on HIV symptoms	4
Lyon, 2013, USA [35]	FACE vs. Treatment as Usual	Adolescent with cancer (14–21) and their Surrogate, dyads *n* = 30 (I: 17, C: 13)	FACE: Family congruence Adolescents decisional conflict Quality of communication	3
Lyon, 2014, USA [36]	FACE vs. Treatment as Usual	Adolescent with cancer (14–21) and their surrogate, dyads *n* = 30 (I: 17, C: 13)	FACE-TC (Family/Adolescent-Centered Advance Care Planning for Teens with Cancer): Satisfaction based on positive and negative experienced emotions (Satisfaction Questionnaire) Quality of life Emotional effects based on anxiety and depression scales Spiritual well-being Participant enrollment, attendance, and retention Data completeness	4

AIDS: acquired immunodeficiency syndrome; HIV: human immunodeficiency virus; * Country where study was conducted.

**Table 3 children-09-00830-t003:** Evidence from observational studies.

Author, Year, Country *	Aim (A), Design (D)	Population (Age in Years), *n*	Outcomes	Risk of Bias Total Score (6)
Friebert, 2020, USA [42]	A: To assess adolescents’ EOL needs and family congruence D: Survey study from intervention arm FACE-TC (FAmily/Adolescent-CEntered Advance Care Planning for Teens with Cancer) (session 1) RCT	Adolescents with cancer (14–21) and their legal or chosen guardian, dyads *n* = 80	FACE-TC Adolescent’s EOL values and needs Family congruence	6
Hays, 2006, USA [37]	A: To assess the effects of DMT (Decision-Making Tool) on family satisfaction and QOL non-experimental pre-test and post-test D: Nonexperimental pre-test, post-test comparison study	Children and adolescents with potentially life-limiting illness (0–22) and their parents, dyads *n* = 41	DMT: Effects on quality of life on four domains (physical, emotional, social, and school functioning) Family satisfaction	4
Hendricks, 2017, USA [38]	A: To evaluate COMPLETE (Communication Plan: Early through End of Life intervention) on the parent and provider levels and to describe the given parental responses. D: Prospective, longitudinal, single-group pilot study	Parents of children (0–18) with a brain tumor and a poor prognosis, mostly mothers; parents *n* = 13 and children *n* = 11	COMPLETE: Parents: emotional well-being (needs, hopes, decision regret, resources, distress, and uncertainty), satisfaction with provider communication and symptom management, and perception of information provided Provider: satisfaction and communication competence	5
Jacobs, 2015, USA [39]	A: To examine EOL family congruence D: Survey study from intervention arm RCT provider post-hoc survey	Adolescents with cancer (14–21) and their legal or chosen guardian, dyads *n* = 17 and clinicians *n* = 30	FACE-TC: Adolescent’s EOL preferences Family congruence Provider survey on three sections: career, FACE-TC interactions, and EOL care experiences	5
Kazmerski, 2016, USA [40]	A: To assess patient and provider attitudes and preferences towards VMC (Voicing My Choices) D: Pre–post-test training survey quality improvement study	Patients with advanced CF (≤22); patients *n* = 12, providers (pre-training) *n* = 6, and providers (post-training) *n* = 7	Patient and provider (pre- and post-training): ACP: positive and negative associations, preferences in CF care VMC: thoughts on VMC and age appropriateness	2
Moody, 2020, USA [41]	A: To assess effects of COMPLETE on EOL outcomes D: Two-phase, single-arm, two-center prospective pre–post-intervention pilot study	Phase I: Parents of children with newly diagnosed cancer (1–<18 months), parents *n* = 21 and children *n* = 18 Phase II: Parents of children with any prognosis, parents *n* = 20 and children *n* = 17	COMPLETE: Parent and child: time of hospice enrollment, pain, EOL interventions, and location of death Parent: negative emotions	4

ACP: advance care planning; CF: cystic fibrosis; EOL: end of life; QOL: quality of life; RCT: randomized controlled trial; * Country where study was conducted.

**Table 4 children-09-00830-t004:** Evidence from mixed-method studies.

Author, Year, Country *	Aim (A), Design (D)	Population (Age in Years), *n*	Outcome Parameters		Risk of Bias Total Score (6)	Quality of Reporting Total Score (32)
			Quantitative	Qualitative		
Kline, 2012, USA [46]	A: To assess family satisfaction and preferences with their palliative care program and its DMT tool (Decision-Making Tool) D: Supportive care survey and open-ended questions interview study	Guardians of high-risk hemato-oncology pediatric patients (mean of 9.7), *n* = 20 (quantitative outcomes) and *n* = 6 (qualitative outcomes)	Understanding treatment options Factors, people and services guiding treatment decisions Effectiveness of the decision-making conference, the palliative care program and DMT	Open-ended questions on the palliative care program and DMT; questions NS	4	6
Lyon, 2019, USA [47]	A: To assess the feasibility and acceptability of FACE-Rare (FAmily-CEntered pediatric Advance Care Planning-Rare) D: Pre–post-test questionnaire study	Pediatric patients with rare diseases (≥1–≤21) and their legal guardians or family caregivers (all mothers), dyads *n* = 6	FACE-Rare Caregiver appraisal Family satisfaction based on positive and negative experienced emotions Families’ quality of communication with providers	Questions NS	5	8.5
Noyes, 2013, UK [50]	A: To evaluate ‘My Choices’ and enhance future care planning D: Pre–post-test questionnaire (quantitative) and semi-structured interview (qualitative) study	Children and young people (0–≥16) with complex health and palliative care needs, as well as their parents and health-care providers, children *n* = 11 parents *n* = 12, bereaved parents *n* = 3, professionals *n* = 13 (qualitative outcomes), professionals (pre-study) *n* = 27, and professionals (post-study) *n* = 20 (quantitative outcomes)	Professionals evaluating My Choices on preferred: Location of care Diverse aspects in palliative care	Views of parents, children, and professionals on the My Choices booklets; questions/themes NS	2	12
Wiener, 2008, USA [49]	A: To assess the acceptability of Five Wishes, helpfulness, and defining important EOL concerns D: Descriptive study data and closed- and open-response interviews	Adolescents and young adults with HIV-1 or metastatic/recurrent cancer (16–28), *n* = 20	Five Wishes: Age appropriateness for someone their age Helpful for someone of the participant’s age Helpful or stressful to the participant	Adjustments to the Five Wishes document	4	11
Wiener, 2012, USA [48]	A: To assess and compare the usefulness, helpfulness, and stressfulness of the MTMWMV (My Thoughts, My Wishes, My Voice) with the Five Wishes D: Descriptive study data and closed- and open-response interviews	AYAs with metastatic or recurrent cancer or HIV infection (16–28), *n* = 52	Evaluating both tools regarding: Age appropriateness for someone their age Helpful for someone of the participant’s age Helpful or stressful to the participant Perceived legality of the document	Adjustments to the MTMWMV document	4	4.5

AYAs: adolescents and young adults; EOL: end of life, HIV: human immunodeficiency virus; NS: not specified; * Country where study was conducted.

**Table 5 children-09-00830-t005:** Evidence from qualitative studies.

Author, Year, Country *	Aim (A), Design (D)	Population (Age in Years), *n*	Outcomes	Quality of Reporting Total Score
Fahner, 2020, the Netherlands [11]	A: To evaluate the acceptability of content of IMPACT (Implementing Pediatric Advance Care Planning Toolkit) D: Qualitative pilot study	Children with life-limiting diseases (0–<18), children *n* = 27, parents *n* = 41, physicians *n* = 11, and nurses *n* = 7	Acceptability of materials Adjustment of tool	8.5
Feraco, 2018, USA [43]	A: To address and ameliorate existing communication gaps in cancer care and to incorporate resulting knowledge in the development of the D100 (the Day 100 talk) D: Qualitative semi-structured interview study	Children, adolescents, and young adults undergoing cancer treatment for from 1 to <7 months (≥13), as well as their parents and oncology providers, adolescents *n* = 5, parents *n* = 6, and providers *n* = 11	Perceived communication gaps in cancer care	18
Finlay, 2008, UK [45]	A: To enhance family engagement in EOL planning through incorporating the results in their 3 × 3 framework D: Documentary analysis study	Children with non-malignant life-limiting illnesses (2–16 months), *n* = 8	Content of EOL plans	4
Hartley, 2016, UK [44]	A: To evaluate the assessment of family needs and concerns by the HNA tool (Holistic Needs Assessment) D: Qualitative analysis study and qualitative pilot study	Care managers employed by Anglia’s Children’s Hospices, *n* = 7	Hopes and reservations Impact on clinical practice Family effect and experiences using the tool Training experiences	10.5

EOL: end of life; * Country where study was conducted.

**Table 6 children-09-00830-t006:** Evidence from descriptive studies.

Author, Year, Country *	Aim (A), Design (D)	Population (Age in Years), *n*	Outcomes	Quality Appraisal
Baker, 2008, USA [58]	A: To assess clinical gaps in pediatric cancer care and to enhance this by integrating these aspects in the tool D: Narrative review study	Children with cancer (NS) and their parents, *n* = NA	The development of the Individualized Care Coordination Plan	NA
Christenson, 2010, USA [51]	A: To present communication gaps in palliative care of adolescents and to improve this by using the CCCT (Comfort Care Communication Tool) D: Case report study	Woman with CF (18), *n* = 1	One case study	NA
Curtin, 2017, USA [52]	A: To assess FACE-TC (FAmily-CEntered pediatric Advance Care Planning-Rare) efficacy on family congruence, quality of life and early ACP document completion D: Study protocol of a dyadic, longitudinal RCT	AYAs (14–20) with cancer and their family decision maker), dyads *n* = 130	Design of dyadic, longitudinal RCT	NA
Dallas, 2012, USA [53]	A: To assess long-term FACE (FAmily/Adolescent-CEntered Advance Care Planning) efficacy on EOL care and tries to enhance physical, psychological, spiritual well-being D: Study protocol of a dyadic, longitudinal RCT	Adolescents with HIV (14–21) and their family decision makers (>21), *n* = 130	Design of dyadic, longitudinal RCT	NA
Fraser, 2010, UK [54]	A: To present the importance of sensitive pediatric EOL planning and to describe the history and format of the Wishes document D: Narrative review study	NA (NS)	The importance of EOL planning The development of the Wishes document	NA
Gallagher, 2018, UK [55]	A: To highlight the importance of knowledge and skills required to engage with children with learning disabilities in their EOL planning D: Narrative review study	NA (NS)	The importance of and challenges in EOL planning ADVANCE toolkit content	NA
Snaman, 2019, USA [59]	A: To identify high-priority factors in cancer treatment decisions and incorporating this in a new tool D: Descriptive study of tool development	AYAs with newly diagnosed high-risk cancers (NS), their parents, and HCPs, dyads *n* = 5 and HCP *n* = 2	Development of MyPref	NA
Toce, 2003, USA [60]	A: To develop a tool that improves the pediatric quality at the EOL D: Descriptive study of tool development	Children with life-threatening conditions (6–>12 months), children *n* = 83 and continuity providers *n* = 105	Development of Footprints	NA
Van Breemen, 2020, Canada [57]	A: To describe the steps in the SICG-peds (Serious illness conversations in pediatrics) using one case as an exampleD: Case report study	Child diagnosed with osteosarcoma (11), *n* = 1	Content of the SICG-Peds	NA
Zadeh, 2015, USA [56]	A: To provide guidelines in the use of Voicing My Choices for health-care providers D: Ethical guide for health-care providers for Voicing My Choices	AYAs living with cancer or pediatric HIV (NS), *n* = NA	Guidelines in the use of Voicing My Choices	NA

ACP: advance care planning; AYAs: adolescents and young adults; CF: cystic fibrosis; EOL: end of life; HCP: health care provider; HIV: human immunodeficiency virus; NS: not specified; NA: not applicable; RCT: randomized controlled trial; * Country where study was conducted.

**Table 7 children-09-00830-t007:** Intervention characteristics.

Intervention (Country)	Intervention Characteristics				Publications Included
	Materials (Ma), Mode (Mo) and Setting (Se)	Aim	Interventionist	Target Population	
1. Comfort Care Communication Tool (USA)	Ma: Four-quadrant design document Mo: Face-to-face longitudinal conversations Se: NS	To enhance adolescents’ disclosure and person-centered care based on families’ goals	Pediatric Advanced Comfort Care Team Nurse	Adolescents with life-threatening or life-limiting health care conditions	Christenson, 2010 [51]
2. Family-Centered pediatric Advance Care Planning (USA)	Ma: Family-centered ACP survey (session 1), Respecting Choices interview (session 2), and Five Wishes document (session 3) Mo: Three-session face-to-face conversation Se: Outpatient clinic	To facilitate EOL discussions for adolescents and their families	Certified facilitator	Adolescents with cancer, HIV or AIDS and their surrogates	Curtin, 2017 [52] Dallas, 2012 [53] Dallas, 2016 [31] Friebert, 2020 [42] Jacobs, 2015 [39] Lyon, 2009 [32] Lyon, 2009 [33] Lyon, 2010 [34] Lyon, 2013 [35] Lyon, 2014 [36]
3. Family-Centered pediatric Advance Care Planning Rare (USA)	Ma: Conversation card, documentation tool Mo: Four-session interviews, face-to-face or via telemedicine conversation Se: NS	To identify and meet caregiver-centered palliative care needs	Certified clinician	Family caregivers of children and adolescents with rare diseases	Lyon, 2019 [47]
4. Implementing Advance Care Planning Toolkit (NL)	Ma: Information leaflets, preparation cards (child and parent), and conversation guides Mo: Face-to-face conversations, on-off conversation, or multiple conversations Se: Home, inpatient, or outpatient clinic	To prepare children, clinicians and parents for future care, to guide documentation, and to elicit the voice of the child and stimulate a patient-centered approach	Clinician involved in the patient’s care	Children with life-limiting conditions and their families	Fahner, 2020 [11]
5. DAY 100 Talk (UK)	Ma: Family preparatory and summary worksheet and a conversation guide Mo: Fill in up-front and face-to-face longitudinal conversations Se: Outpatient clinic	To enhance families’ disclosure and interdisciplinary guidance	Trained pediatric oncologist and psychosocial clinician	Children, adolescents, and young adults with cancer and their families	Feraco, 2018 [43]
6. 3 × 3 Lifetime Framework (UK)	Ma: 3 × 3 Framework Document Mo: Face-to-face longitudinal conversations Se: NS	To enhance family engagement in EOL planning	Clinicians	Children with non-malignant, life-limiting illnesses and their families	Finlay, 2008 [45]
7. The Wishes Document (UK)	Ma: Hand-held document Mo: Face-to-face longitudinal conversations Se: NS	To enhance family engagement in EOL planning	Clinician involved in the patient’s care	Children, young people with life-limiting conditions and their families	Fraser, 2010 [54]
8. The ADVANCE toolkit (UK)	Ma: Ethical guide Mo: Face-to-face longitudinal conversations Se: Private place	To enhance provider guidance, families’ disclosure, and families’ engagement in EOL planning	Clinician involved in the patient’s care	Young persons with learning disabilities (who are approaching the end of life) and their families	Gallagher, 2018 [55]
9. Holistic Needs Assessment (UK)	Ma: Comprehensive assessment of needs Mo: Face-to-face conversation Se: NS	To enhance person-centered care based on family needs	Senior member of staff	Children in palliative care settings and their family	Hartley, 2016 [44]
10. Decision-making Communication Tool (USA)	Ma: Four domains of decision making Mo: Face-to-face longitudinal conversations Se: Outpatient clinic	To enhance patient–provider communication, decision making, and quality of life, as well as to identify goals of care	Supportive care team clinicians	Pediatric palliative care: infants, children, and adolescents with potentially life-limiting illnesses (oncology) and their families	Kline, 2012 [46] Hays, 2006 [37]
11.Communication Plan: Early through End of Life (USA)	Ma: Conversation guide and visual aids Mo: Three face-to-face conversation sessions, longitudinal revision Se: During clinic appointments	To reduce parental distress	Trained oncology providers	Parents of children with cancer	Hendricks, 2017 [38] Moody, 2020 [41]
12. Voicing my choices (USA)	Ma: Guide adapted from the Five Wishes, completion of the document guide Mo: Longitudinal revision Se: NS	To enhance communication between the patient and caregiver in EOL preferences and care	Clinicians	Adolescents and young people living with a serious illness	Wiener, 2012 [48] Kazmerski, 2016 [40] Zadeh, 2015 [56]
13. My Choices/Choices for My Child Booklets (UK)	Ma: Booklets for children and parents, possibility Mo: To fill in/initiate thinking or face-to-face conversations Se: Home or outpatient clinic	To enhance family engagement in future planning and the disclosure of family preferences	NA	Children with life-limiting conditions from diagnosis onwards and their parents	Noyes, 2013 [50]
14. The Serious Illness Conversation Guide-Peds (SICG-Peds) (Canada)	Ma: Conversation guide Mo: Longitudinal face-to-face or by phone conversations Se: Home or clinic	To enhance understanding of illness and care preferences	Trained pediatrician	Children with serious illness and their parents	Van Breemen, 2020 [57]
15. Five Wishes^®^ (USA)	Ma: Legal document consisting of five wishes Mo: Fill in document Se: NS	To enhance communication in EOL care	Clinicians	Adolescents and young adults living with serious illnesses	Wiener, 2008 [49]
16.Individualized care planning and coordination (USA)	Ma: Advance care planning documentation tool Mo: Longitudinal revision on timely basis Se: NS	To facilitate integration of palliative care into ongoing care	Clinicians	Children with cancer and their parents	Baker, 2008 [58]
17. MyPref (USA)	Ma: Preference report up-front cancer therapy Mo: Fill in document, longitudinal revision Se: NS	To clarify AYAs’ preferences and to enhance engagement in medical decision making	Oncology providers or other clinicians	AYA patients with relapsed/progressive cancer	Snaman, 2019 [59]
18. FOOTPRINTS (USA)	Ma: Conversation guide, using a discharge order sheet Mo: Longitudinal face-to-face conversations Se: During the interdisciplinary “care conference”	To provide quality of care for the patient, their families, and providers through anticipating their needs on a continual basis	Hospital-based “continuity” pediatrician	Children with life-limiting illnesses and their families	Toce, 2003 [60]

AYA: adolescents and young adults; ACP: advance care planning; AIDS: acquired immunodeficiency syndrome; EOL: end of life; HIV: human immunodeficiency virus; NA: not applicable; NS: not specified.

**Table 8 children-09-00830-t008:** Age-appropriate characteristics.

Article	Description Concept	Implementation in the Tool Described		Evaluation on Age Appropriateness Stated by Patient/Provider/Family			Recommendations
		Statement of Concept Applied	Elements of Tool	Patient	Provider	Family	
Baker, 2008 [58]	Implicit	No	NS	NS	NS	NS	Yes
Christenson, 2010 [51]	Implicit	Yes	Questions adjusted for age and maturity	NS	NS	NS	Yes
Curtin, 2017 [52]	Implicit	Yes	NS	NS	NS	NS	No
Dallas, 2012 [53]	Implicit	Yes	NS	NS	NS	NS	No
Dallas, 2016 [31]	No Description	No	NS	NS	NS	NS	No
Fahner, 2020 [11]	Explicit	Yes	Booklets and conversation guides, with language adapted to the children	NS	NS	NS	Yes
Feraco, 2018 [43]	No Description	No	NS	NS	NS	NS	No
Finlay, 2008 [45]	No Description	No	NS	NS	NS	NS	No
Fraser, 2010 [54]	Implicit	No	NS	NS	NS	NS	Yes
Friebert, 2020 [42]	Implicit	No	NS	NS	NS	NS	No
Gallagher,2018 [55]	Implicit	No	NS	NS	NS	NS	Yes
Hartley, 2016 [44]	Implicit	No	NS	NS	NS	NS	No
Hays, 2006 [37]	Implicit	No	NS	NS	NS	NS	Yes
Hendricks,2017 [38]	No Description	No	NS	NS	NS	NS	No
Jacobs, 2015 [39]	No description	No	NS	NS	NS	NS	No
Kazmerski, 2016 [40]	Implicit	No	NS	90% considered VMC (Voicing My Choices) to be age-appropriate; 66% considered ACP to be appropriate to introduce before the age of 18 or at any age	58% considered VMC to be appropriate for patient population/age group; 50% found the ideal patient age for ACP discussion was >18 years	NS	No
Kline, 2012 [46]	No Description	No	NS	NS	NS	NS	No
Lyon, 2009 [32]	Explicit	Yes	NS	NS	NS	NS	Yes
Lyon, 2009 [33]	Explicit	Yes	NS	NS	NS	NS	No
Lyon, 2010 [34]	No Description	No	NS	NS	NS	NS	No
Lyon, 2013 [35]	Implicit	No	NS	NS	NS	NS	No
Lyon, 2014 [36]	Implicit	Yes	NS	NS	NS	NS	No
Lyon, 2019 [47]	No Description	No	NS	NS	NS	NS	No
Moody, 2020 [41]	No Description	No	NS	NS	NS	NS	No
Noyes, 2013 [50]	Implicit	Yes	Booklets content and images adapted for age	NS	NS	NS	Yes
Snaman, 2019 [59]	Explicit	Yes	NS	NS	NS	NS	Yes
Van Breemen, 2020 [57]	Implicit	Yes	Family-centered language	NS	NS	NS	No
Wiener, 2008 [49]	Implicit	Yes	Age-appropriate images	90% declared that all statements on EOL care were appropriate and helpful for someone their age	NS	NS	Yes
Wiener, 2012 [48]	Implicit	Yes	Wording and questions adjusted for development and a glossary added	No significant tool differences in the degree of help or stress in age groups or differences in document content; AYAs disagreed on whether medical care wishes in the Five Wishes versus MTMWMV (My Thoughts, My Wishes, My Voice) was more appropriate for someone of their age	NS	NS	Yes
Zadeh, 2015 [56]	Explicit	Yes	Wording and questions adjusted for development and a glossary added	NS	NS	NS	Yes
Toce, 2003 [60]	Implicit	Yes	NS	NS	NS	NS	No

ACP: advance care planning; AYA: adolescents and young adults; EOL: end of life; NS: not specified.

**Table 9 children-09-00830-t009:** Factors related to age appropriateness.

	Willingness to Participate	Ability to Participate			Developing Social Identity	Legal Responsibilities
		Decision-Making Capacity	A Child’s Understanding of Their Own Medical Process	Cognitive Impairment		
Baker, 2008 [58]		x	x			
Christenson, 2010 [51]	x	x	x		x	
Curtin, 2017 [52]				x		
Dallas, 2012 [53]	x	x	x	x		x
Dallas, 2016 [31]	x			x		x
Fahner, 2020 [11]	x	x				
Feraco, 2018 [43]	x			x		
Finlay, 2008 [45]						
Fraser, 2010 [54]		x				
Friebert, 2020 [42]	x	x	x	x		
Gallagher, 2018 [55]		x	x		x	
Hartley, 2016 [44]		x				
Hay, 2006 [37]						
Hendricks, 2017 [38]						x
Jacobs, 2015 [39]	x			x		x
Kazmerski, 2016 [40]	x					
Kline, 2012 [46]						
Lyon, 2009 [32]	x	x		x	x	
Lyon, 2009 [33]	x	x	x	x		x
Lyon, 2010 [34]	x			x		x
Lyon, 2013 [35]	x	x	x	x		
Lyon, 2014 [36]	x	x	x	x		
Lyon, 2019 [47]						
Moody, 2020 [41]						
Noyes, 2013 [50]						
Snaman, 2019 [59]	x		x			
Toce, 2003 [60]		x				
van Breemen, 2020 [57]			x			
Wiener, 2008 [49]		x	x			x
Wiener, 2012 [48]	x	x	x		x	x
Zadeh, 2015 [56]	x	x	x		x	x

## Data Availability

Not applicable.

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
