# Peer review of "Age-Appropriate Advance Care Planning in Children Diagnosed with a Life-Limiting Condition: A Systematic Review"

_children, 2022, doi:10.3390/children9060830_

Round 1
Reviewer 1 Report
This systematic review of age-appropriate ACP in children with life-limiting conditions has been very well conducted and is a worthy addition to the literature in children's palliative care.
However, there are many errors in grammar and syntax, particularly in the Discussion, that need to be carefully resolved to make it fit for publication and not detract from the information presented.
I would like to see the Authors make comment on the heavy USA weighting (23 of 31 articles) of this literature which could be considered a bias. Importantly, just over 25% (6 of 23 articles) of the USA derived data are from one Author, Lyon et al.
The Authors have correctly discussed the insufficient evaluation of tools for age-appropriateness and commented on the majority of articles making an "implicit" reference to the age-appropriate concept without linking the importance of tool adaptation to development of the child. I suggest the Authors could go further by discussing the lack of validation of tools and how this could impact on usefulness. There could be more discussion on the three articles which did evaluate their tools for age-appropriateness.
Given this was a thorough review I would challenge the Authors to "rank" the tools for age-appropriateness and discuss "Next Steps" to improve the available tools.
Reviewer 2 Report
The authors took on a critically important topic; age appropriate advanced care planning. There are a growing number of pediatric ACP tools available, but figuring out which is appropriate for which developmental age is challenging.
It is an important contribution but could be strengthened with the following
1) The first paragraph of the introduction should bring in the concept of developmentally appropriate involvement (rather than waiting til the second) - it is a critically important part of the analysis/application
2) For the methods, some more details would be helpful
- How they defined Pediatric ACP
- What specific criteria were used to narrow from the abstracts to full text (and who did that)
3) For the results, additional details would be helpful
- Where the study was conducted (not entirely sure if the country was country of publication, where the study was, or what). If there were any differences by country/region
- If there was racial/ethnic variation in the participants (and what the implications are)
4) For the results, the tables are large and bulky, making the hard to follow. Wondering what can be combined (potentially using landscaped tables). Lots in Table 4 and 5 might lend itself to a bar group. Also wondering about something like a waterfall plot visually showing which ages each study covered - could very nicely highlight the preponderance of AYA studies/interventions.
